# Caregiver Survey-Based Perspectives on Digital Therapeutics for Children with Delayed Language Development

**DOI:** 10.3390/healthcare13243290

**Published:** 2025-12-15

**Authors:** Jinju Lee, Sejin Kwon, Jin Young Ko, Yulhyun Park, Jaewon Lee, Ju Seok Ryu, Seo Yeon Yoon, Jee Hyun Suh

**Affiliations:** 1Department of Rehabilitation Medicine, Seoul National University Bundang Hospital, Seongnam 13620, Republic of Korea; jjlee4@snu.ac.kr (J.L.); rq300@snubh.org (S.K.); rq045@snubh.org (J.L.); jseok337@daum.net (J.S.R.); 2Department of Rehabilitation, Gyeonggi Provincial Medical Center Anseong, Anseong 17568, Republic of Korea; gpmc7296@medical.or.kr; 3Department of Rehabilitation, Gyeonggi Provincial Medical Center Icheon, Icheon 17366, Republic of Korea; yulhyunp@gmail.com; 4Department and Research Institute of Rehabilitation Medicine, Yonsei University College of Medicine, Seoul 03722, Republic of Korea; seoyeon0521@gmail.com

**Keywords:** delayed language development, articulation therapy, digital therapeutics, caregiver survey, UTAUT2, speech-language pathology, mobile health

## Abstract

**Objective:** This study aimed to identify caregivers’ perceptions, preferences, and intentions regarding the use of digital articulation therapy applications for children with DLD. **Methods:** A cross-sectional online survey was conducted between August and September 2025 among 197 caregivers of children diagnosed with DLD. A 43-item questionnaire was structured into five domains addressing demographics, therapy experience, digital-therapy exposure, and preferences for application-based articulation therapy. Constructs from the Unified Theory of Acceptance and Use of Technology 2, including performance expectancy (PE), social influence (SI), price value (PV), facilitating conditions (FC), and behavioral intention, were analyzed. **Results:** Most respondents were mothers (96%), and 78.7% of them resided in urban areas. Among them, 84.3% had prior speech therapy experience. Only 15.7% had used educational or therapeutic applications; the main dissatisfaction factors were lack of fun (51.6%) and feedback (19.3%). Caregivers preferred short, engaging sessions (20–40 min per day), video-based (75%) or game-based (64%) content, and feedback every 2 months, with a reasonable monthly cost (20,000–30,000 KRW). Regression analysis revealed that sex (β = −0.451, *p* = 0.013), PE (β = 0.381, *p* < 0.001), and PV (β = 0.212, *p* = 0.034) significantly associated with behavioral intention to use an articulation-therapy application, whereas SI and FC were not significant. **Conclusions:** Caregivers of children with DLD demonstrated strong willingness to adopt digital articulation therapy applications, particularly when these tools provide meaningful therapeutic outcomes at moderate cost and include motivating, interactive content. Future application design should prioritize treatment functionality, user engagement, and accessibility to enhance adoption and continuity of digital speech-language interventions.

## 1. Introduction

Delayed language development (DLD) is a common condition with prevalence estimates between 3 and 10% across different populations [1]. Prior studies indicate that the prevalence is approximately 7% in the United States and 8.5% in China [2,3]. Later in life, DLD may lead to emotional disorders, behavioral problems, learning disabilities, school dropout, social maladjustment, and other psychiatric conditions [4,5]. Therefore, early identification and timely intervention at a young age are crucial to ensure adequate treatment.

Conventional speech therapy is usually provided in person through one-on-one sessions with a speech-language therapist in a hospital or a clinical environment. The cost per individual one-on-one speech-language therapy session is estimated to range between $75 (lower estimate) and $149 (upper estimate) [6]. For children with DLD who receive therapy once or twice weekly, this represents a substantial financial burden. In addition to these therapeutic expenses, there are other direct costs, such as regular physician visits and fees for language assessment. In the United States, DLD accounts for an estimated 2.5–3% of the nation’s gross domestic product each year, corresponding to approximately $154–186 billion [7]. Moreover, indirect costs arise when parents must adjust their work schedules or leave the workforce entirely to provide necessary care, resulting in a reduction in household income [8]. Additionally, conventional speech-language therapy results in unavoidable disparities in accessibility and service availability between urban and rural areas [6]. There are marked geographical disparities not only in the number of hospitals that provide speech-language therapy and in the availability of qualified speech-language therapists but also in the accessibility of public transportation to these treatment facilities, which varies considerably across regions [9,10].

To overcome these limitations, supplementary tools, such as digital therapeutics for speech-language therapy, have been developed. However, owing to several constraints, including limited applicability across different disorder types, difficulty in maintaining children’s engagement and long-term adherence, and challenges in objectively measuring therapeutic effectiveness, these tools have not been widely adopted or found to be satisfactory by children with DLD and their caregivers [11,12]. The present survey was conducted to identify the needs and preferences of caregivers of children with DLD, with the aim of developing a digital therapeutic solution tailored towards this population.

## 2. Materials and Methods

### 2.1. Study Design and Participants

This cross-sectional survey was conducted between August 2025 and September 2025 among caregivers of children with DLD. Data were collected using an online questionnaire designed to explore the caregivers’ characteristics and perceptions. Inclusion criteria were as follows: (1) caregivers of children aged between 12 months and 18 years who had been diagnosed with DLD; (2) individuals who accessed and completed the online survey using an Internet-connected device; and (3) those who fully understood the purpose and methods of the study and voluntarily agreed to participate. Exclusion criteria was caregivers with linguistic or cognitive limitations that made it difficult for them to understand or respond to the survey. The target sample size was set at 200 caregivers, referencing a recent caregiver survey study in Children (2025) that recruited 180 participants and assuming an approximate 10% drop-out rate [13]. In addition, the sample size satisfies regression-based recommendations, as Green’s (1991) guideline (N ≥ 50 + 8 m) requires at least 106 participants for seven variables [14]. The final sample of 197 caregivers therefore met both the target recruitment and statistical adequacy criteria required for the study’s aims.

### 2.2. Survey Design

The study team developed an online survey involving three physiatrists and one speech-language pathologist, based on a review of previous literature on caregivers of children with DLD. The final questionnaire comprised 43 items organized into five domains: (1) demographic characteristics of caregivers; (2) characteristics of children with DLD; (3) current status of speech therapy and other rehabilitative therapies; (4) experiences and environment related to application-based language therapy; and (5) caregivers’ preferences regarding digital articulation therapy delivered through mobile applications (Appendix A). Some items were designed as conditional questions that appeared only when a respondent selected specific answers to preceding questions. The survey was distributed through a secure online platform, and estimated time to complete the questionnaire was 10–15 min. The participants submitted their responses only once to prevent duplicate entries.

We adopted a user-experience-centered approach similar to that of the Unified Theory of Acceptance and Use of Technology 2 (UTAUT2), recognizing the importance of users’ subjective experiences. To explore satisfaction according to prior technology-use patterns, the questionnaire included items assessing participants’ previous experiences with similar digital services and their satisfaction with such technologies. Additionally, the survey examined caregivers’ preferences regarding specific features of articulation therapy applications, including preferred content types, desired feedback frequency, and maximum amount they were willing to pay for the service.

### 2.3. Study Procedure and Ethical Considerations

The study protocol was reviewed and approved by the Institutional Review Board of our institution (approval number: B-2507-987-303). All the participants were informed of the purpose, content, and voluntary nature of the study prior to their participation. Electronic informed consent was obtained prior to completion of the survey. The survey was conducted anonymously, and all collected data were kept confidential and used only for study purposes.

### 2.4. Analytical Framework

This study adopted five key constructs from the Unified Theory of Acceptance and UTAUT2: performance expectancy (PE), social influence (SI), price value (PV), facilitating conditions (FC), and behavioral intention (intention) (Appendix A). Some UTAUT2 constructs were adapted to more precisely reflect the context of home-based speech digital therapeutic application while maintaining their original conceptual meaning. For instance, FC were operationalized as the maximum amount of time that caregivers could allocate to home-based articulation practice, because time availability represents the most salient resource constraint in this setting. This context-specific operationalization allows the construct to remain theoretically consistent with UTAUT2 while enhancing its practical relevance and interpretability for caregivers. Each construct was assessed using a context-specific, single-item indicator developed for this study. PE refers to an expected outcome of technology use. This study was operationalized as the anticipated benefit of the application-based language therapy. SI represents the degree to which individuals perceive how important others recommend the technology. Five response options were included: recommendations from professionals, brand recognition, online parenting communities, experienced users, and social media marketing. PV indicates perceived reasonableness of the cost associated with using the technology. In this study, the maximum monthly amount that caregivers were willing to pay for app-based language therapy was calculated. FC refers to the environmental and resource-related factors that enable technology use. Four items assessed whether caregivers could support the application use by their child, maximum time allowed for home-based speech therapy, availability of a quiet space, and ownership of a tablet personal computer (PC). Behavioral intention (intention) was defined as willingness to use technology. This was measured using an item assessing the caregiver’s intention to use a language therapy application while waiting for in-person therapy services. As this study focused on pre-use perceptions of digital therapeutic applications, the constructs of hedonic motivation and habit from the original UTAUT2 model were not included.

### 2.5. Statistical Analysis

Descriptive statistics were used to analyze frequency and percentage of responses for each item. In this study, 197 out of 200 participants completed the survey, and only these complete responses were included in the analysis. As all questionnaire items were set as mandatory in the online survey system, no item-level missing data were present. Therefore, no imputation procedures were required, and a complete-case analysis was conducted. Prior to the main analysis, the distributions of the variables were inspected using histograms and skewness values to identify potential outliers; however, the evaluation of regression assumptions was based on the residuals. For regression analysis, Q-Q plots were examined to assess the normality of the residuals, which indicated that the residuals were approximately normally distributed. The Pearson’s correlation analysis was performed to assess the relationship among variables and evaluate multicollinearity. Finally, a multiple regression analysis was conducted to examine the effects of UTAUT2 constructs on the intention to use an articulation therapy application. Each construct was measured using a single item adapted from the core components of the validated UTAUT2 scale. Statistical significance was set at *p* < 0.05. All statistical analyses were conducted using IBM SPSS Statistics software (version 29.0; IBM Corp., Chicago, IL, USA).

## 3. Results

### 3.1. Demographic Information of Responders

In total, 197 caregivers of children with DLD participated in the survey.

Most respondents were mothers and the majority resided in urban areas, including the Seoul metropolitan area (Seoul, Gyeonggi-do, and Incheon) (Table 1). The children were typically preschool- or early school-age. Nearly half of the households were dual-income, and daytime childcare was most commonly provided by the caregiver, followed by daycare centers of kindergartens. Disabilities were noted for approximately one-quarter of the children, with autism spectrum disorder and speech/language disability being the most frequent. In addition, most children had prior experience with speech therapy (Table 1).

### 3.2. Speech and Rehabilitation Therapy Experiences

Among caregivers whose children had experience in speech therapy, 33.7% had also received other types of rehabilitation therapy, whereas none of the caregivers whose children had experience in speech therapy experience participated in any other rehabilitation therapy. The institutions and types of rehabilitation therapies (excluding speech therapy) reported by respondents were as follows:-Among caregivers whose children received additional rehabilitation therapies, 55 reported that their child has underwent ≥3 different types of rehabilitation therapy.-Private centers were the most common type of institution providing these therapies (73.2%), with each child receiving an average of 1.32 types of therapy. This was followed by tertiary/university hospitals (57.1%), with an average of 2.56 therapies per child, clinics (33.9%) with 1.47 therapies, general hospitals (25%) with 2.07 therapies, and welfare centers (21.4%) with 1.25 therapies. In summary, more than half of the caregivers reported that their children received costly, non-certified therapy services, such as those provided by private therapy centers or home-visit therapy programs.-Among the 166 caregivers whose children had experience in speech therapy, 85.5% reported that their child had been undergoing therapy for >1 year, and 79.5% received therapy at least twice per week. One-way travel time to the therapy institution was typically 30–60 min or longer.

### 3.3. Experience with Application-Based Educational or Therapeutic Services

Only a small proportion of respondents reported having experience using application-based educational or therapeutic services for their child (Table 2).

Among them, average satisfaction level was seven of nine, indicating that most users were generally satisfied with their experience. Among respondents whose child had used application-based educational or therapeutic services, the most frequently reported dissatisfaction factor was “lack of fun”, followed by “lack of feedback”. Additional comments included difficulties finding therapeutic options suitable for their children within the application. When analyzed by the child’s disability type, caregivers of children with physical disabilities (N = 8) mostly cited “lack of feedback” (50%) and “high cost” (37.5%) as primary sources of dissatisfaction. Among caregivers of children with autism spectrum disorder (N = 19), “lack of fun” (78.9%) was the most frequently reported dissatisfaction factor. Similarly, in the language disorder group (N = 19), “lack of fun” (73.7%) was the predominant complaint.

### 3.4. Preference for the Articulation Therapy Application

The overall responses to digital therapeutics are shown in Table 3. First, the preference for conducting speech therapy through media devices, such as mobile phones or tablet PCs, was measured using a 9-point Likert scale, where higher scores indicated greater preference. Respondents demonstrated a generally positive attitude toward media-based articulation therapy application (Table 3). Second, among respondents, a session length of approximately 20 to <40 min was most commonly preferred, whereas very short durations were least preferred. When analyzed by disability type, caregivers of children with brain lesions (n = 4), hearing impairment (n = 8), and intellectual disability (n = 5) mostly selected “≥1 h” (100%, 50%, and 60%, respectively). Third, in the context of articulation therapy through an application, the item assessing the relative importance of assessment versus treatment was measured using a bipolar scale, where values closer to 1 indicated greater emphasis on assessment, and values closer to 9 indicated greater emphasis on treatment. Responses ranged between 3 and 9, indicating that caregivers generally prioritized treatment over assessment (Table 3). Fourth, in this study, the items about preferred content, attitudes toward game-based content, and feedback frequency were collected only from potential consumers who had no prior experience with educational or therapeutic applications but expressed a willingness to use an articulation therapy application during the waiting period for speech therapy. The preferred content item allowed multiple responses, with “video-based content” being the most frequently selected, followed by “game-based (quiz-type) content”. Preference for game-based content was measured using a 9-point Likert scale, with higher scores indicating greater preference. The scores ranged between 3 and 9, indicating a generally positive attitude toward game-based elements in articulation therapy applications. The most preferred feedback interval for articulation therapy applications was “every 2 months”, while “every 3 months” was the least selected. When analyzed by disability type, caregivers of children with autism spectrum disorder (n = 6) and language disorder (n = 5) most frequently selected “every 1.5 months” (66.7% and 60%, respectively), whereas those with hearing impairment (n = 7) most commonly choose “every 1 month” (57.1%). Finally, results regarding the prerequisites for using an articulation therapy application were collected as multiple responses. Among all the respondents, “accuracy of diagnosis and treatment” was the most frequently selected requirement, while “fun” was the least selected. When analyzed by disability type, caregivers of children with autism spectrum disorder (n = 26), language disorder (n = 24), and brain lesion (n = 4) most frequently selected “reasonable cost” as the most important prerequisite (84.6%, 75%, and 100%, respectively). In contrast, caregivers of children with hearing impairment (n = 8) most frequently selected “ease of application use” (87.5%).

### 3.5. UTAUT-2 Model

(1)Perceived PE of Speech Therapy

Among all the respondents, each of them selected an average of 2.16 desired outcomes from speech therapy (multiple responses allowed). Specifically, 20.8% of the respondents selected one desired outcome, 52.3% selected two, 1.5% selected three, and 10.2% selected four, suggesting that expectations for speech therapy are multifaceted. The most frequently chosen goals were “improvement in communication difficulties” (79.2%) and “accurate pronunciation” (72.6%). When analyzed according to disability type, caregivers of children with physical disabilities (N = 8) most often selected “accurate pronunciation” (60%) as their primary goal, whereas caregivers of children with all other disability types most commonly chose “improvement in communication difficulties” as the main expected outcome of speech therapy.

(2)Perceived SI in Articulation Therapy Application Usage

Among the multiple factors considered important for using an articulation therapy application, the most frequently selected item was “recommendation from exports” (72.1%). However, when analyzed by disability type, caregivers of children with autism spectrum disorder (n = 19) and language disorder (n = 19) selected “brand recognition” (69.2% and 66.7%, respectively) as the most important factor.

(3)PV of the Articulation Therapy Application

In this study, responses to the maximum willingness-to-pay question were collected only from potential consumers who had no prior experience with educational or therapeutic applications but expressed willingness to use an articulation therapy application during the waiting period for speech therapy. Among these respondents, the most frequently selected monthly maximum payment range for articulation therapy application was 20,000–30,000 KRW, whereas the proportion willing to pay ≥40,000 KRW was the lowest (Table 3).

(4)FC of Articulation Therapy Application Us

FC comprised four items: perceived parental role, time available, available home space, and ownership of a tablet PC. First, when asked about parental assistance during home-based application use for speech therapy, the most frequently selected response was “I should assist the child, and I am able to do so” (72.1%), while “I am unable to assist and believe it is better not to” (1.5%) was selected the least. By disability type, among children with physical disabilities (n = 10), the most frequent response was “I am able to assist, but I believe it is better not to” (50%). For children with language disorders (n = 24) and autism spectrum disorders (n = 26), the most common response was “I should assist the child, but it is difficult for me to do so.” Second, regarding the maximum amount of time that could be allocated for home-based speech therapy, the most frequent response was “1 to <2 h” (34%), whereas “<1 h” (7.6%) was the least common. By disability type, the most common responses were “2 to <3 h” among those with physical disabilities (N = 10, 50%), brain lesion (n = N, 100%), and intellectual disabilities (N = 5, 60%), while children with hearing impairment (N = 8) most frequently selected “≥3 h” (50%). Finally, 98% of the respondents reported having a quiet space suitable for speech therapy at home, and all the disability groups indicated that such a space was available in their households. Additionally, 85.3% of the respondents stated that they owned a tablet PC suitable for conducting speech therapy at home.

(5)Intention to Use the Articulation Therapy Application

Responses regarding willingness to use an articulation-therapy application during the waiting period for speech therapy were collected only from individuals with no prior experience using educational or therapeutic applications. Willingness was measured using a 9-point Likert scale, with scores between 1 and 9 (M = 6.43, SD = 1.703). For further analysis, the responses were categorized into two groups: low (1–4 points) and high (5–9 points) willingness. Overall, 89.2% of respondents fell into the high willingness category. When analyzed according to registered disability type, all the caregivers of children with brain lesion, hearing impairment, language disorder, and intellectual disability indicated high willingness to use the application (100%), followed by those with autism spectrum disorder (85.7%) and physical disabilities (50%). Additionally, among respondents who indicated low willingness to use the application, the most frequently cited reason was “aversion to using digital medial” (33.3%), while “no perceived need for use” (5.6%) was the least reported reason.

(6)Intention to Use an Articulation Digital Therapeutic Application Based on the UTAUT2 Model

In this study, the effects of PE, SI, FC, and PV on intention were examined using regression analysis. All the variables were measured using a single-item scale (Table 4). In this survey, PE was defined as the perceived outcome of speech therapy, comprising four selectable items: Improvement in Communication Difficulties, Accurate Pronunciation, Better Interaction with Peers, and Increased Confidence. These items were converted into a 1–4-point scale, where higher scores indicated a greater desire for diverse performance outcomes. SI refers to the important social factors associated with caregiver’s use of articulation therapy application, comprising five selectable items: Brand Awareness, Recommendation by Experts, Recommendation from online Communities, Social Media Marketing, and Recommendation by Other users. PV comprised five price categories: <10,000 KRW, ≥10,000 KRW, ≥20,000 KRW, ≥30,000 KRW, and ≥40,000 KRW. This variable was also converted to a 1–5-point scale, where higher scores represented greater price acceptability. In this study, FC was measured as the maximum amount of time that caregivers could allocate to home-based speech therapy for their children and was converted into a 1–4-point scale, with higher scores indicating a greater amount of time available for speech therapy at home. Intention was measured using a 9-point Likert scale assessing the willingness to use the application during the waiting period for speech therapy. Higher scores indicate a stronger intention to use the articulation therapy application.

The normality of regression residuals was confirmed using Q-Q plots, and Pearson’s correlation analysis was conducted. The results showed significant correlations between the intention to use an articulation therapy application during the waiting period and sex (r = −0.198, *p* < 0.05), speech therapy experience (r = 0.386, *p* < 0.01), PE (r = 0.352, *p* < 0.01), SI (r = 0.217, *p* < 0.01), and PV (r = 0.303, *p* < 0.01). Based on the Cohen’s criteria [15] (0.10 = small, 0.30 = medium, and 0.50 = large), speech therapy experience, PE, and PV demonstrated moderate correlations with intention, whereas sex and SI showed weak correlations. Additionally, no multicollinearity concerns were identified, as none of the correlation coefficients exceeded r = 0.80. Variance inflation factor (VIF) and tolerance value were also confirmed. All VIF values ranged from 1.028 to 1.437, and all tolerance values exceeded 0.696, indicating no concerns regarding multicollinearity (Appendix A). Results of the multiple regression analysis revealed that sex, PE, and PV significantly were associated with intention to use the application during the waiting period (Figure 1). Specifically, male children showed a 0.451-point lower intention to use the application than females (*p* = 0.013). A higher number of expected outcomes from speech therapy (PE) was associated with a 0.381-point increase in intention (*p* < 0.001), whereas a higher PV was associated with a 0.212-point increase in intention (*p* = 0.034). These findings indicate that greater performance expectancy and higher PV, as well as being female, are associated with a higher intention to use articulation therapy applications during the waiting period (Figure 1).

## 4. Discussion

This caregiver survey conducted among caregivers of children with DLD aimed to explore the perspectives on digital therapeutics to guide the development of speech therapy applications and digital therapeutic tools that meet consumer needs and are acceptable to both children with DLD and their caregivers. The findings revealed that caregivers preferred applications offering video- and game-based contents designed to engage children’s interest, with a daily usage time of 20–40 min and a monthly cost of approximately 20,000–30,000 KRW. Additionally, caregivers indicated that a feedback interval of once every 2 months was considered appropriate following the use of such speech therapy applications.

This study investigated caregivers’ willingness to use digital therapeutics to treat DLD. In this study, 197 of 200 participants completed the survey, resulting in a response rate of approximately 98.5%. This high participation rate suggests a strong willingness among caregivers to engage in studies on digital therapeutic solutions for DLD [12]. In this study, 96% of the respondents were mothers. This high proportion of female respondents is consistent with prior evidence suggesting that women are more likely to seek healthcare services than men [16]. Previous studies revealed that 66.9% of speech language pathologists participating in surveys were females [12,17], indicating that sex may be related to individuals’ participation in studies on communication disorder. Studies indicate that women tend to seek medical services more frequently than men, a pattern that could extend to study participation [12,18]. Additionally, women often assume caregiving roles, leading to greater involvement in their children’s health-related activities [12,19,20,21].

A consistent explanation for caregivers’ positive attitudes toward fun and engagement may lie in the central role that motivation plays in home-based speech therapy for young children. Prior studies have shown that gamified elements enhance children’s willingness to participate and sustain their attention during therapeutic tasks. For example, previous study reported that digital games effectively increase children’s motivation and participation in language rehabilitation [22]. Likewise, Virag et al. found that more than half of caregivers believed that digital applications can enhance children’s motivation and interest, facilitating continued speech-therapy practice at home [23]. These findings align with the responses observed in the present study. Caregivers viewed game-based features favorably, and “lack of fun” emerged as the most frequently reported limitation of previously used applications. Taken together, these results suggest that enjoyable and engaging components are not merely desirable but are perceived—both through prior experience and parental expectations—as essential for sustaining children’s participation and promoting continued use of digital articulation-therapy applications. Based on the UTAUT-2 model, sex, PE, and PV significantly explained the intention to use an articulation therapy application during the speech therapy waiting period. In particular, a significant association was observed between sex and intention to use the application. Specifically, caregivers of female children demonstrated a higher intention to adopt digital therapeutic applications than caregivers of male children. The association between child sex and behavioral intention reflects the higher intention reported among caregivers of girls compared to boys. Although UTAUT2 does not focus on child characteristics, developmental literature suggests that girls often show higher compliance, greater sustained attention, and stronger engagement in structured learning activities than boys [24,25]. These behavioral tendencies may lead caregivers to view digital articulation therapy applications as more suitable or effective for girls, thereby increasing their willingness to adopt such tools. Our results demonstrated that caregivers who expected more diverse outcomes from speech therapy exhibited a higher intention to use the application (B = 0.381, *p* < 0.001). Therefore, emphasizing therapeutic goals, particularly improving communication abilities and achieving accurate speech production, may enhance caregivers’ willingness to adopt articulation therapy applications. Additionally, a higher acceptance of cost was associated with a greater intention to use the application (B = 0.121, *p* < 0.05). PV typically increases when users perceive that the benefits of a technology outweigh its cost [26], suggesting that pricing aligned with the perceived therapeutic value of digital therapeutics may support broader caregiver acceptance and utilization.

In contrast, SI was not significantly associated with the intention to use the application in this study. This finding aligns with a meta-analysis of the UTAUT-2 model by Tamilmani et al. [27]; they reported that SI generally plays a limited role in technology adoption, with its effect varying according to technology type. However, the non-significant effect of SI in this study may also reflect differences in operationalization compared with traditional UTAUT-2 measures. In prior UTAUT-2 study, SI typically refers to the extent to which individuals perceive how important others believe that they should use a technology [12]. In comparison, this study measured SI as the number of socially relevant factors that may be associated with application use, including brand awareness, recommendation by experts, recommendation from online communities, social media marketing, and recommendation by other users. Although prior studies have found SI to be non-significant in certain technological contexts, such as mobile banking [27], little is known about how SI relates to the adoption of digital therapeutic applications. Moreover, specific sources of influence, such as clinicians, peers, prior users, and brand recognition, may have different patterns of association with caregivers’ adoption of digital therapeutics. Another possible explanation for the non-significant effect of SI is that decisions regarding children’s therapy may be driven more by caregivers’ personal evaluations of their child’s needs rather than by external social expectations. In the context of pediatric speech therapy, caregivers often assume primary responsibility and may rely on their own judgment when considering home-based digital interventions. This interpretation is consistent with evidence from parental decision-making research showing that caregivers often prioritize their child’s well-being, family values, and individualized goals, and are regarded as “experts in their child’s care” when navigating therapeutic decisions for children with complex needs [28]. Therefore, future studies should examine the nuanced roles of different social and the differentiated roles of various social influence pathways in promoting the uptake of digital speech-language therapy applications.

Similarly, FC was not significantly associated with intention to use the articulation therapy application. In previous studies, FC refers to the extent to which individuals perceive that organizational, technical, and environmental resources and support are available to enable the use of technology [12]. In contrast, the present study operationalized FC as the maximum amount of time caregivers could allocate to assisting their children with digital therapeutics at home. Although time availability represents a meaningful and relevant resource in the context of home-based digital therapy, it may not fully capture the broader structural supports—such as technical infrastructure, device accessibility, or environmental stability—reflected in the original UTAUT2 framework. This more context-specific operationalization may have constrained the variability and reduced the predictive utility of FC in this study. Future studies should explore FC more directly in a digital therapy context, including availability of suitable devices, internet access, a child’s attention capacity, and level of support needed during application use. Clarifying the environmental and resource factors that meaningfully support application adoption and sustained use is critical in improving real-world implementation and long-term engagement with therapy applications.

According to previous studies, approximately 60% of patients with communication disorders and their caregivers prefer a combination of face-to-face visits and mobile applications rather than either modality alone [16]. Therefore, it is essential to consider how digital therapeutics can be effectively integrated with in-person therapy to establish a harmonious hybrid treatment model. Building on this context, the present findings offer several implications for the design of future digital therapeutic platforms for children with DLD. First, the strong effects of performance expectancy and price value indicate that caregivers place considerable importance on perceivable therapeutic benefits and reasonable costs. Digital therapeutic applications should therefore prioritize transparent communication of expected outcomes, provide evidence-based progress tracking, and offer pricing models that caregivers perceive as affordable and worthwhile. Second, engaging, game-based features are not merely optional but essential for sustaining children’s participation in home-based therapy. Incorporating developmentally appropriate reward cycles, interactive tasks, and visually appealing interfaces may enhance motivation and adherence. Third, the non-significance of social influence and facilitating conditions in our study may suggest that caregivers rely more on their personal judgments rather than external opinions or structural resources when deciding whether to adopt digital therapy tools. As such, developers should emphasize individualized onboarding, clear instructional guidance, and simplified session workflows rather than expecting social recommendation pathways to drive adoption.

### Limitations

This study has several limitations. First, although the sample size of 197 caregivers was sufficient for the study’s analytical goals, the demographic composition was not fully representative; most respondents were mothers (96%) and urban residents (78.7%), indicating a sampling bias that may limit generalizability. Future research should recruit more diverse caregiver groups, including fathers and caregivers from rural regions, to better capture the full spectrum of perspectives. Second, because the study only included Korean participants, the generalizability of the findings to the global population is limited. However, this restriction was necessary because of the need to analyze the unique phonetic characteristics of the Korean language and associated articulation errors. Third, the use of single-item measures may introduce greater measurement error than multi-item scales. Because each UTAUT2 construct in this study was assessed with a single item, item–factor correlations, internal consistency indices, and other psychometric evaluations could not be calculated. Although this limitation precluded the examination of internal consistency reliability, prior methodological literature indicates that single-item measures can be appropriate when constructs are conceptually narrow, easily understood, and when the analytical focus is on associations rather than latent variable validation [29,30,31]. In the present study, the aim was not to validate the full UTAUT2 measurement model but to examine how the key constructs relate to caregivers’ behavioral intention to use a digital therapeutic application. Given this study purpose, the use of single-item indicators is unlikely to compromise the interpretability of the findings. Nevertheless, future research may employ multi-item scales to strengthen measurement validity in the context of digital therapeutic applications. Fourth, the generalizability of the results is limited because only 15.7% of participants had prior experience with digital therapy apps. Their limited familiarity may have shaped their attitudes and intentions differently from users who have greater exposure. Future research should include caregivers with more diverse levels of experience. Finally future studies should include a larger and more geographically diverse sample encompassing both urban and rural populations across South Korea to enhance the external validity of the findings.

## 5. Conclusions

Taken together, these findings suggest that an articulation therapy application for children should be designed to support short, manageable daily sessions (approximately 20–40 min per day) at a moderate monthly cost (approximately 20,000–30,000 KRW) and include video- or game-based contents to enhance engagement and motivation. Applications that emphasize therapeutic intervention over assessment and provide periodic feedback every 2 months may be particularly effective and acceptable for families. Based on these insights, future developments should focus on creating digital applications that promote language development in children with DLD.

## Figures and Tables

**Figure 1 healthcare-13-03290-f001:**
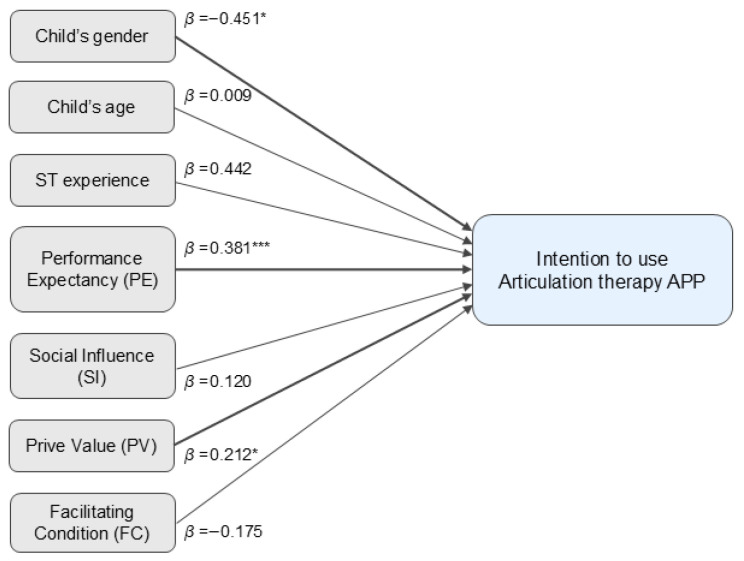
Intention to Use an Articulation Therapy Application According to the Unified Theory of Acceptance and Use of Technology 2 Model. * *p* < 0.05; *** *p* < 0.001.

**Table 1 healthcare-13-03290-t001:** Demographic data.

Characteristics (Total = 197)	N (%)
Relationship with the Child	
Mother	190 (96%)
Father	5 (3%)
Relative	2 (1%)
Sibling	0
Grandparent	0
Region of the Child’s Residence	
Urban	155 (78.7%)
Rural	22 (11.2%)
Child’s gender	
Boy	122 (62%)
Girl	75 (38%)
Child’s current age Mean (SD), range	5.5 (SD 1.621), 2–11
Dual-income Family	
Yes	86 (43.7%)
No	111 (56.3%)
Primary Daytime Caregiver of the Child	
Self (Respondent)	79 (40.1%)
Co-residing Family Member (Other than Respondent)	30 (15.2%)
Relative (Non-residing or Extended Family)	12 (6.1%)
Home-visit Childcare Provider (or In-home Babysitter)	1 (0.5%)
Daycare Center/Kindergarten	75 (38.1%)
Comorbidity	
Hearing difficulty	42 (21.3%)
History of Frenectomy	49 (24.9%)
Disability Registration	54 (27.4%)
Autism Spectrum Disorder	26 (13.2%)
Speech/Language Disability	24 (12.2%)
Hearing Disability	8 (4.1%)
Intellectual Disability	5 (2.5%)
Brain lesion	4 (2.0%)
Physical Disability	10 (5.1%)
Speech Therapy Experience	166 (84.3%)
Speech Therapy Duration	
Less than 1 year	24 (14.5%)
1–2 years	106 (63.9%)
2–3 years	25 (15.1%)
3–4 years	10 (6%)
More than 4 years	1 (0.6%)
Frequency of Speech Therapy	
Once per week	34 (20.5%)
Twice per week	80 (48.2%)
Three times per week	45 (27.1%)
Four times per week	3 (1.8%)
Five or more times per week	4 (2.4%)
Waiting Period for Speech Therapy	
No waiting	77 (39.1%)
Less than 1 year	96 (48.7%)
1–2 years	20 (10.2%)
2–3 years	3 (1.5%)
3–4 years	1 (0.5%)
More than 4 years	0
Type of Speech Therapy Institution	
Tertiary general hospital/university hospital	47 (26.1%)
General hospital	16 (8.9%)
Local clinic (primary care)	12 (6.7%)
Private therapy center	96 (53.3%)
Home-visit therapy	1 (0.6%)
Welfare center	7 (3.9%)
Special education school	0
Daycare center/kindergarten	1 (0.6%)
Travel Time to Therapy Location	
Less than 30 min	52 (26.4%)
30–60 min	131 (66.5%)
60–90 min	12 (6.1%)
90–120 min	2 (1%)
More than 120 min	0
Expected Duration of Continued Speech Therapy	
Less than 1 year	4 (2%)
1–2 years	49 (24.9%)
2–3 years	87 (44.2%)
3–4 years	32 (16.2%)
More than 4 years	25 (12.7%)

**Table 2 healthcare-13-03290-t002:** Application-Based Education/Therapy Experience.

Questions	N (%)
Having Application-Based Education/Therapy Experience	31 (15.74%)
Intention to Use an Articulation Therapy Application During the Waiting Period for Institutional Speech Therapy (caregivers of children without experience using Education/Therapy Application, N = 166)	6.43 (SD = 1.703)
**Questions (Total = 31)**	**N (%)**
User Satisfaction	7.03 (SD = 0.482)
Dissatisfaction with the Application Use	
Lack of fun	16 (51.6%)
Low Accuracy	5 (16.1%)
High Cost	3 (9.7%)
Lack of Feedback	6 (19.3%)
No Particular Dissatisfaction/None	0
Insufficient Personalization of Therapy (Other Comment)	1 (3.2%)

**Table 3 healthcare-13-03290-t003:** Attitudes Toward Digital Therapeutics.

Questions	N (%)
Preference for Conducting Speech Therapy via Digital Media (e.g., Smartphone, Tablet PC)	6.27 (SD = 1.49)
Perceived Maximum Daily Duration (in Minutes) for Media-Based Speech Therapy for Children	
Less than 20 min	6 (3%)
20 to less than 40 min	77 (39.1%)
40 to less than 60 min	70 (35.5%)
1 h or more	44 (22.3%)
Perceived Importance Between Assessment and Treatment in Application-Based Articulation Therapy	6.84 (SD = 1.248)
3	1 (0.55%)
4	5 (2.5%)
5	22 (11.2%)
6	47 (23.9%)
7	69 (35%)
8	32 (16.2%)
9	21 (10.7%)
Prerequisites for Using an Application for Articulation Therapy	
Appropriateness of Cost	92 (46.7%)
Accuracy of Diagnosis and Therapy	114 (57.9%)
Ease of Use	63 (32%)
User Customization	68 (34.5%)
Accuracy of Feedback	34 (17.3%)
Fun	2 (1%)
Accreditation by Professional Organizations	12 (6.1%)
Important Factors in Selecting and Using an Application for Articulation Therapy	
Brand Awareness	77 (39.1%)
Recommendation by Experts	142 (72.1%)
Recommendation from Online Communities	55 (27.9%)
Social Media Marketing	9 (4.6%)
Recommendation by Other Users	50 (25.4%)
Preference for Application Content and Game-Based Features	6.47 (SD = 1.280)
Feedback Frequency (Potential Users Without Prior Application Experience but Willing to Use It During the Waiting Period, N = 148)	
Every 1 month	48 (32.4%)
Every 1.5 months	20 (13.5%)
Every 2 months	61 (41.2%)
Every 2.5 months	15 (10.1%)
Every 3 months	4 (2.7%)
Maximum Willingness to Pay per Month for Application-Based Articulation Therapy (Potential Users Without Prior Application Experience but Willing to Use It During the Waiting Period, N = 148)	
Less than 10,000 KRW	16 (10.8%)
10,000–19,999 KRW	21 (14.2%)
20,000–29,999 KRW	74 (50%)
30,000–39,999 KRW	24 (16.2%)
40,000 KRW or more	13 (8.8%)

**Table 4 healthcare-13-03290-t004:** Descriptive Statistics for the Measurement Model (Mean SD).

Construct	Number of Items	Item	Mean ± SD
Performance Expectancy	1	Perceived Outcomes of Speech Therapy (1–4 Scale)	2.16 ± 0.87
Social Influence	1	Social factors in Using the Articulation Therapy Application (1–5 Scale)	1.69 ± 0.46
Facilitating Conditions	1	Maximum Time Available for Home-Based Speech Therapy for the Child (1–4 Scale)	2.79 ± 0.94
Price Value	1	Price Acceptability for the Articulation Therapy Application (1–5 Scale)	2.98 ± 1.05
Intention	1	Intention to Use the Articulation Therapy Application During the Waiting Period (1–9 Scale)	6.43 ± 1.70

## Data Availability

The data are not publicly available due to ethical and privacy restrictions involving patient information. Data may be available from the corresponding author upon reasonable request.

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
