# Peer review of "Caregiver Survey-Based Perspectives on Digital Therapeutics for Children with Delayed Language Development"

_healthcare, 2025, doi:10.3390/healthcare13243290_

Round 1

Reviewer 1 Report

Comments and Suggestions for Authors

This is a timely study.

Well-written manuscript. Some suggestions to improve the clarity and flow are the following:

  • The authors are being inconsistent when referring to the participants, alternating between caregivers, children, and subgroups without re-anchoring (e.g., “among the 166 children…” vs. “among respondents…”).

  • Consider mentioning which disability or issue you are referring to because it's hard for the reader to follow the subgroups and analysis

  • Avoid reiterating the same findings that are in the tables, especially in the descriptive results section
  • Sometimes authors refer to the technology as application-based therapy and sometimes as digital therapeutics. I think some consistency is needed here.
  • I think there should be an explanation to why your participants had that attitude towards fun and engagement
  • Some constructs of the UTAUT were operationalized differently than the standard UTAUT-2. Some sentences should be added to explain that.
  •  It is noteworthy that your analysis should not be referring to findings in a context of causal association because it's just association analysis

Author Response

Healthcare, Review of Caregiver Survey-Based Perspectives on Digital Therapeutics for Children with Delayed Language Development

Dear Editor:

Please consider our revised manuscript, “Caregiver Survey-Based Perspectives on Digital Therapeutics for Children with Delayed Language Development”, for publication in the Healthcare. 

We appreciate the interest that the editors and reviewers have taken in our manuscript and the constructive criticism they have given. We have addressed the major concerns of the reviewers. We have also included a point-by-point response to the reviewers in addition to making the changes as described in the manuscript. Changes to the text in the manuscript are marked in red.

Our main findings remain unchanged: Caregivers of children with DLD are highly willing to adopt digital articulation therapy applications, especially when they offer meaningful therapeutic benefits at a reasonable cost and include engaging, interactive content.

Thank you again for consideration of our revised manuscript.

Sincerely yours,

Jee Hyun Suh, M.D., Ph.D.

Department of Rehabilitation Medicine, Seoul National University Bundang Hospital, Seoul National University College of Medicine, 82 Gumi-ro 173 Beon-gil, Bundang-gu, Seongnam-si, Gyeonggi-do, South Korea, 13620

Phone: 82-31-787-7741

E-mail: jeehyun.suh1@gmail.com

Reviewer 1.

  1. The authors are being inconsistent when referring to the participants, alternating between caregivers, children, and subgroups without re-anchoring (e.g., “among the 166 children...” vs. “among respondents...”).

Answer> Thank you for pointing out the inconsistency in how participants were referenced throughout the manuscript. We have thoroughly revised the text to ensure that the primary unit of analysis—caregivers—is described consistently across all sections. Mentions of children are now clearly framed as characteristics reported by the caregivers (e.g., “caregivers of children with …”), and subgroup descriptions have been rewritten to maintain a consistent anchor and avoid ambiguity. These revisions enhance clarity and ensure uniform terminology throughout the manuscript.

  1. Consider mentioning which disability or issue you are referring to because it's hard for the reader to follow the subgroups and analysis

Answer> Thank you for pointing this out. We have revised the Results section to explicitly identify the disability type in each subgroup analysis (e.g., autism spectrum disorder, physical disabilities, language disorders) to ensure clarity and improve readability.

(line 198-202): When analyzed by the child’s disability type, caregivers of children with physical disabilities (N=8) mostly cited “lack of feedback” (50%) and “high cost” (37.5%) as primary sources of dissatisfaction. Among caregivers of children with autism spectrum disorder (N=19), “lack of fun” (78.9%) was the most frequently reported dissatisfaction factor. Similarly, in the language disorder group (N=19), “lack of fun” (73.7%) was the predominant complaint.

(line 230-233): When analyzed by disability type, caregivers of children with autism spectrum disorder (n=6) and language disorder (n=5) most frequently selected “every 1.5 months” (66.7% and 60%, respectively), whereas those with hearing impairment (n=7) most commonly choose “every 1 month” (57.1%).

(line 236-240): When analyzed by disability type, caregivers of children with autism spectrum disorder (n=26), language disorder (n=24), and brain lesion (n=4) most frequently selected “rea-sonable cost” as the most important prerequisite (84.6%, 75%, and 100%, respectively). In contrast, caregivers of children with hearing impairment (n=8) most frequently selected “ease of application use” (87.5%).

(line 255-259): Among the multiple factors considered important for using an articulation therapy application, the most frequently selected item was “recommendation from exports” (72.1%). However, when analyzed by disability type, caregivers of children with autism spectrum disorder (n=19) and language disorder (n=19) selected “brand recognition” (69.2% and 66.7%, respectively) as the most important factor.

(line 273-277): By disability type, among children with physical disabilities (n=10), the most frequent re-sponse was “I am able to assist, but I believe it is better not to” (50%). For children with language disorders (n=24) and autism spectrum disorders (n=26), the most common re-sponse was “I should assist the child, but it is difficult for me to do so.”

(line 293-297): When analyzed according to registered disability type, all the caregivers of children with brain lesion, hearing impairment, language disorder, and intellectual disability indicated high willingness to use the application (100%), followed by those with autism spectrum disorder (85.7%) and physical disabilities (50%).

  1. Avoid reiterating the same findings that are in the tables, especially in the descriptive results section.

Answer> Thank you for the suggestion. We revised the descriptive results section to avoid repeating numerical values already presented in the tables. The text now summarizes key patterns and highlights overall trends, while detailed figures are retained in the tables for clarity.

  1. Sometimes authors refer to the technology as application-based therapy and sometimes as digital therapeutics. I think some consistency is needed here.

Answer> Thank you for pointing out the inconsistency in terminology. We have clarified the terminology and revised the manuscript to ensure consistent use. Specifically, we now use the term digital therapeutic application throughout the manuscript, except when referencing prior studies or when describing the broader category of digital therapeutics as a field. This revision improves conceptual clarity and maintains consistency in how the technology is described

  1. I think there should be an explanation to why your participants had that attitude towards fun and engagement

Answer> Thank you for the insightful comment. We agree that additional explanation is needed regarding participants’ attitudes toward fun and engagement. In the revised manuscript, we contextualized these findings by discussing how caregivers’ preferences may reflect their expectations for children’s sustained participation, motivation, and compliance during home-based articulation therapy. Given that digital therapeutic tools for children often rely on gamified or engaging elements to maintain attention, caregivers may view fun and engagement as essential components for successful therapy. This explanation has been added to the Discussion section

(lines 367–380): A consistent explanation for caregivers’ positive attitudes toward fun and engage-ment may lie in the central role that motivation plays in home-based speech therapy for young children. Prior studies have shown that gamified elements enhance children’s willingness to participate and sustain their attention during therapeutic tasks. For exam-ple, previous study reported that digital games effectively increase children’s motivation and participation in language rehabilitation [21]. Likewise, Virag et al. found that more than half of caregivers believed that digital applications can enhance children’s motiva-tion and interest, facilitating continued speech-therapy practice at home [22]. These find-ings align with the responses observed in the present study. Caregivers viewed game-based features favorably, and “lack of fun” emerged as the most frequently reported limitation of previously used applications. Taken together, these results suggest that en-joyable and engaging components are not merely desirable but are perceived—both through prior experience and parental expectations—as essential for sustaining children’s participation and promoting continued use of digital articulation-therapy applications.

  1. Some constructs of the UTAUT were operationalized differently than the standard UTAUT-2. Some sentences should be added to explain that.

Answer> Thank you for pointing this out. In addition to using single-item indicators, some UTAUT2 constructs were intentionally operationalized in a context-specific manner to better reflect the characteristics of home-based speech digital therapeutic application. For example, facilitating conditions were adapted to represent the maximum amount of time caregivers could allocate to home-based speech digital therapeutic application, as time availability is a critical resource in this setting. Although these context-specific operationalizations differ from the standard UTAUT2 scale structure, they preserve the conceptual meaning of each construct while improving relevance and interpretability for the target population. We have added clarifications to the Methods sections.  

(lines 116–124): Some UTAUT2 constructs were adapted to more precisely reflect the context of home-based speech digital therapeutic application while maintaining their original conceptual mean-ing. For instance, FC were operationalized as the maximum amount of time that caregivers could allocate to home-based articulation practice, because time availability represents the most salient resource constraint in this setting. This context-specific operationalization allows the construct to remain theoretically consistent with UTAUT2 while enhancing its practical relevance and interpretability for caregivers. Each construct was assessed using a context-specific, single-item indicator developed for this study.

  1. It is noteworthy that your analysis should not be referring to findings in a context of causal association because it's just association analysis.

Answer> Thank you for pointing this out. We fully agree that the cross-sectional design and regression analysis used in this study cannot infer causality. Accordingly, we have revised the manuscript to avoid any causal language and consistently describe the results in terms of associations rather than causal effects. Terms such as “influence,” “affect,” or “predict” have been replaced with non-causal expressions such as “were associated with” or “be related to” These revisions have been applied throughout the Results and Discussion sections.

(line 310-313): SI refers to the important social factors associated with caregiver’s use of articulation therapy application, comprising five selectable items: Brand Awareness, Recommendation by Experts, Recommendation from online Communities, Social Media Marketing, and Recommendation by Other users.

(line 332-334): Results of the multiple regression analysis revealed that sex, PE, and PV significantly were associated with intention to use the application during the waiting period (Table 6).

(line 404-405): In contrast, SI was not significantly associated with the intention to use the application in this study.

(line 411-414): In comparison, this study measured SI as the number of socially relevant factors that may be associated with application use, including brand awareness, recommendation by experts, recommendation from online communities, social media marketing, and recommendation by other users.

(line 427-428): Similarly, FC was not significantly associated with intention to use the articulation therapy application.

Reviewer 2 Report

Comments and Suggestions for Authors

The manuscript presents a investigation into digital articulation therapy adoption among caregivers of children with developmental language disorder (DLD). The use of UTAUT2 constructs, caregiver-centered insights, and regression modeling provide valuable evidence for digital therapeutics development. The study is well designed and clearly reported; however, revision is requested as per following comments,

  1. Provide reliability metrics (Cronbach’s alpha) for all UTAUT2 constructs to confirm internal consistency.
  2. Clarify whether multicollinearity diagnostics were performed (VIF, tolerance values).
  3. Specify whether normality assumptions for regression residuals were tested.
  4. Provide more detail on how missing data, if any, were handled.
  5. Clarify sample size justification (e.g., power analysis or UTAUT2 guidelines for modeling).
  6. Expand interpretation on why SI and FC were not significant, especially given SI's established role in UTAUT2.
  7. Provide deeper explanation of the negative association between sex and behavioral intention, and discuss potential cultural or demographic factors.
  8. Strengthen the linkage between results and design implications for future digital therapeutic platforms.
  9. Report item-factor correlations or provide a supplementary table summarizing UTAUT2 item performance.
  10. Clarify if caregivers piloted the survey prior to launch for comprehension testing.
  11. Acknowledge sampling bias, as the sample is overwhelmingly composed of mothers (96%) and urban residents (78.7%).
  12. Discuss limited generalizability due to low prior exposure to digital therapy apps (15.7%).
  1. The following studies are suggested to evaluate and add to the literature review of the manuscript: https://doi.org/10.1371/journal.pone.0242023, https://doi.org/10.1016/j.ejphar.2024.176759, https://doi.org/10.3390/jtaer20030231

Author Response

Healthcare, Review of Caregiver Survey-Based Perspectives on Digital Therapeutics for Children with Delayed Language Development

Dear Editor:

Please consider our revised manuscript, “Caregiver Survey-Based Perspectives on Digital Therapeutics for Children with Delayed Language Development”, for publication in the Healthcare. 

We appreciate the interest that the editors and reviewers have taken in our manuscript and the constructive criticism they have given. We have addressed the major concerns of the reviewers. We have also included a point-by-point response to the reviewers in addition to making the changes as described in the manuscript. Changes to the text in the manuscript are marked in red.

Our main findings remain unchanged: Caregivers of children with DLD are highly willing to adopt digital articulation therapy applications, especially when they offer meaningful therapeutic benefits at a reasonable cost and include engaging, interactive content.

Thank you again for consideration of our revised manuscript.

Sincerely yours,

Jee Hyun Suh, M.D., Ph.D.

Department of Rehabilitation Medicine, Seoul National University Bundang Hospital, Seoul National University College of Medicine, 82 Gumi-ro 173 Beon-gil, Bundang-gu, Seongnam-si, Gyeonggi-do, South Korea, 13620

Phone: 82-31-787-7741

E-mail: jeehyun.suh1@gmail.com

  1. Provide reliability metrics (Cronbach’s alpha) for all UTAUT2 constructs to

confirm internal consistency.

Answer> Thank you for the comment. The four constructs (PE, SE, PV, FC) were measured using single-item indicators that were derived from the core items of the validated UTAUT2 scale. Although multi-item scales are commonly used to measure UTAUT2 constructs, previous literature has also suggested that single-item measures can be appropriate when constructs are concrete, unidimensional, and easily understood by respondents, or when the primary purpose of the study is prediction rather than latent variable validation. In this study, each construct represents a clearly defined and familiar concept for caregivers, and the goal of the analysis was not to test or refine the UTAUT2 measurement model, but rather to examine whether these key perceptions influence caregivers’ behavioral intention to use a digital therapeutic application. Thus, using single-item measures was considered appropriate to:

  • reduce respondent burden among caregivers with limited time,
  • minimize survey fatigue,
  • maximize completion rates
  • directly capture the central evaluative meaning of each construct.

Therefore, internal consistency metrics (e.g., Cronbach’s alpha) could not be calculated, as they require multiple items; instead, reliability was addressed through evidence-based item development grounded in prior UTAUT2 research and expert review to ensure conceptual clarity and appropriateness. This clarification has been added to the Limitation sections.

(line 472-485): Third, the use of single-item measures may introduce greater measurement error than mul-ti-item scales. Because each UTAUT2 construct in this study was assessed with a single item, item–factor correlations, internal consistency indices, and other psychometric evalu-ations could not be calculated. Although this limitation precluded the examination of in-ternal consistency reliability, prior methodological literature indicates that single-item measures can be appropriate when constructs are conceptually narrow, easily understood, and when the analytical focus is on associations rather than latent variable validation [28-30]. In the present study, the aim was not to validate the full UTAUT2 measurement model but to examine how the key constructs relate to caregivers’ behavioral intention to use a digital therapeutic application. Given this study purpose, the use of single-item in-dicators is unlikely to compromise the interpretability of the findings. Nevertheless, future research may employ multi-item scales to strengthen measurement validity in the context of digital therapeutic applications.

  1. Clarify whether multicollinearity diagnostics were performed (VIF, tolerance values).

Answer> Multicollinearity diagnostics were performed as part of the regression analysis. Specially, variance inflation factor(VIF) and tolerance values were calculated for all predictors. The results indicated no issues of multicollinearity, with all VIF values falling well below the commonly accepted threshold. These results have been added to the manuscript text, and the full set of VIF and tolerance values is provided in Supplementary Table 3 and the manuscript.

(line 330-332): Variance inflation factor (VIF) and tolerance value were also confirmed. All VIF values ranged from 1.028 to 1.437, and all tolerance values exceeded 0.696, indicating no con-cerns regarding multicollinearity (Table S3).

  1. Specify whether normality assumptions for regression residuals were tested.

Answer> We appreciate the reviewer’s valuable comment. In the revised manuscript, we have clarified how the normality assumption for the regression analysis was evaluated. Prior to the main analysis, the distributions of the variables were inspected using histograms and skewness values to identify potential outliers; however, regression assumptions were assessed based on the residuals. Specifically, Q-Q plots of the regression residuals were examined, and the residuals were found to be approximately normally distributed. This clarification has been added to the Methods section.

 (line 147-151): Prior to the main analysis, the distributions of the variables were inspected using histo-grams and skewness values to identify potential outliers; however, the evaluation of re-gression assumptions was based on the residuals. For regression analysis, Q-Q plots were examined to assess the normality of the residuals, which indicated that the residuals were approximately normally distributed.

  1. Provide more detail on how missing data, if any, were handled.

Answer> Thank you for this helpful suggestion. In this study, 197 out of 200 participants completed the survey, resulting in a response rate of approximately 98.5%. Only complete survey responses were included in the analysis; therefore, no item-level missing data were present. The three participants who provided only partial responses were excluded from the analysis, and thus no imputation or listwise deletion was necessary. This clarification has been added to the Methods section

(lines 142–145): In this study, 197 out of 200 participants completed the survey, and only these complete responses were included in the analysis. As all questionnaire items were set as mandatory in the online survey system, no item-level missing data were present. Therefore, no imputation procedures were required, and a complete-case analysis was conducted.

  1. Clarify sample size justification (e.g., power analysis or UTAUT2 guidelines for

modeling).

Answer> Thank you for this important comment. We have added a clear justification for the sample size in the revised manuscript (Section 2.1). Specifically, the target sample size of 200 caregivers was determined with reference to a recent study published in Children (Basel) (2025), which successfully recruited 180 caregivers in a similar pediatric rehabilitation survey. Considering an anticipated 10% drop-out rate, a target recruitment of 200 participants was established. In addition, because the primary analysis involved multiple regression rather than structural equation modeling, we evaluated statistical adequacy using regression-based guidelines. According to Green’s (1991) criterion (N ≥ 50 + 8m), a minimum of 106 participants is required when seven predictors are included. Our final sample of 197 caregivers exceeded both the field-based target sample and the recommended minimum for regression analysis. This sample size also provides sufficient power to detect medium effect sizes (Cohen, 1988). The manuscript has been updated to reflect these justifications.

(lines 79-84): The target sample size was set at 200 caregivers, referencing a recent caregiver survey study in Children (2025) that recruited 180 participants and assuming an approximate 10% drop-out rate[13]. In addition, the sample size satisfies regression-based recommendations, as Green’s (1991) guideline (N ≥ 50 + 8m) requires at least 106 participants for seven variables[14]. The final sample of 197 caregivers therefore met both the target recruitment and statistical adequacy criteria required for the study’s aims.

  1. Expand interpretation on why SI and FC were not significant, especially given SI's established role in UTAUT2.

Answer> Thank you for this insightful comment. We have expanded the interpretation of the non-significant effects of social influence (SI) and facilitating conditions (FC). In the revised Discussion, we explain that SI may have been less influential in this context because caregivers often make decisions about children’s therapy based on personal judgment and the child’s needs, rather than external opinions. Additionally, FC was measured as the amount of time caregivers could allocate to home-based practice; however, time availability alone may not fully capture the broader structural supports typically represented by FC in UTAUT2. These contextual factors may explain why SI and FC did not emerge as significant predictors in this study. The expanded explanation is now included in the Discussion section

(line 418-426):  Another possible explanation for the non-significant effect of SI is that decisions regarding children’s therapy may be driven more by caregivers’ personal evaluations of their child’s needs rather than by external social expectations. In the context of pediatric speech therapy, caregivers often assume primary responsibility and may rely on their own judgment when considering home-based digital interventions. This interpretation is consistent with evidence from parental decision-making research showing that caregivers often prioritize their child’s well-being, family values, and individualized goals, and are regarded as “experts in their child’s care” when navigating therapeutic decisions for children with complex needs.

(line 435-440): Although time availability represents a meaningful and relevant resource in the context of home-based digital therapy, it may not fully capture the broader structural sup-ports—such as technical infrastructure, device accessibility, or environmental stabil-ity—reflected in the original UTAUT2 framework. This more context-specific operational-ization may have constrained the variability and reduced the predictive utility of FC in this study.

  1. Provide deeper explanation of the negative association between sex and behavioral intention, and discuss potential cultural or demographic factors.

Answer> Thank you for pointing this out. We have expanded our interpretation of the negative association between child sex and behavioral intention. In our study, the negative coefficient indicates that caregivers of girls exhibited higher intention to use the articulation therapy application compared to caregivers of boys. Prior research suggests that girls tend to show higher compliance, sustained attention, and willingness to participate in structured learning activities, which may lead caregivers to perceive digital therapeutic applications as more suitable or effective for them. The revised explanation has been added to the Discussion section.

(lines 388–394): The association between child sex and behavioral intention reflects the higher intention reported among caregivers of girls compared to boys. Although UTAUT2 does not focus on child characteristics, developmental literature suggests that girls often show higher com-pliance, greater sustained attention, and stronger engagement in structured learning activ-ities than boys [23, 24]. These behavioral tendencies may lead caregivers to view digital ar-ticulation therapy applications as more suitable or effective for girls, thereby increasing their willingness to adopt such tools.

8.Strengthen the linkage between results and design implications for future digital therapeutic platforms.

Answer> Thank you for this valuable suggestion. We have strengthened the connection between our empirical findings and the design implications for future digital therapeutic platforms. In the revised Discussion, we elaborate on how key predictors—such as performance expectancy, price value, social influence, facilitating conditions and child sex—should inform user-centered design strategies. Specifically, we highlight the importance of incorporating engaging, game-based elements to enhance children’s motivation, ensuring value-aligned pricing models for caregivers, and providing clear evidence of therapeutic effectiveness. We also discuss how non-significant predictors (e.g., social influence, facilitating conditions) suggest the need for personalized guidance rather than reliance on external social pressure. These implications have been integrated into the revised manuscript.

 (lines 450–461): Building on this context, the present findings offer several implications for the design of future digital therapeutic platforms for children with DLD. First, the strong effects of per-formance expectancy and price value indicate that caregivers place considerable im-portance on perceivable therapeutic benefits and reasonable costs. Digital therapeutic ap-plications should therefore prioritize transparent communication of expected outcomes, provide evidence-based progress tracking, and offer pricing models that caregivers per-ceive as affordable and worthwhile. Second, engaging, game-based features are not merely optional but essential for sustaining children’s participation in home-based therapy. In-corporating developmentally appropriate reward cycles, interactive tasks, and visually appealing interfaces may enhance motivation and adherence. Third, the non-significance of social influence and facilitating conditions in our study may suggest that caregivers rely more on their personal judgments rather than external opinions or structural resources when deciding whether to adopt digital therapy tools. As such, developers should em-phasize individualized onboarding, clear instructional guidance, and simplified session workflows rather than expecting social recommendation pathways to drive adoption.

9.Report item-factor correlations or provide a supplementary table summarizing

UTAUT2 item performance.

Answer>  Thank you for this valuable comment. We agree that reporting item–factor correlations or providing a supplementary table is typically appropriate for multi-item scales. However, in the present study, each UTAUT2 construct was measured using a single-item indicator, and therefore, item–factor correlations, factor loadings, or internal consistency indices cannot be computed. This single-item approach was chosen intentionally to minimize respondent burden among caregivers and because the study’s analytical goal was prediction-oriented rather than focused on validating the latent measurement structure of UTAUT2. To address the reviewer’s concern, we have clarified this rationale in the revised manuscript.

(line 451-482): Third, the use of single-item measures may introduce greater measurement error than mul-ti-item scales. Because each UTAUT2 construct in this study was assessed with a single item, item–factor correlations, internal consistency indices, and other psychometric evalu-ations could not be calculated. Although this limitation precluded the examination of in-ternal consistency reliability, prior methodological literature indicates that single-item measures can be appropriate when constructs are conceptually narrow, easily understood, and when the analytical focus is on associations rather than latent variable validation.

  1. Clarify if caregivers piloted the survey prior to launch for comprehension testing.

Answer> Thank you for this question. A formal pilot test with caregivers was not conducted prior to survey launch. However, all items were adapted from well-established UTAUT2 constructs and were intentionally phrased as single, concise statements to minimize the risk of comprehension difficulties. Because the study targeted busy caregivers, our priority was ensuring a high response and completion rate within a limited data collection window. Notably, the final dataset showed a near-perfect completion rate with no missing responses, suggesting that caregivers did not experience difficulties in understanding the items.

11.Acknowledge sampling bias, as the sample is overwhelmingly composed of mothers (96%) and urban residents (78.7%).

Answer> Thank you for pointing this out. We agree that the sample composition reflects a notable demographic concentration, with the majority of respondents being mothers (96%) and residing in urban areas (78.7%). We have now explicitly acknowledged this sampling bias in the revised manuscript and noted that these characteristics may limit the generalizability of the findings to more diverse caregiver populations, including fathers and families living in rural regions. This limitation has been added to the Discussion section.

(lines 464-469): , although the sample size of 197 caregivers was sufficient for the study’s analytical goals, the demographic composition was not fully representative; most respondents were moth-ers (96%) and urban residents (78.7%), indicating a sampling bias that may limit general-izability. Future research should recruit more diverse caregiver groups, including fathers and caregivers from rural regions, to better capture the full spectrum of perspectives.

  1. Discuss limited generalizability due to low prior exposure to digital therapy apps (15.7%).

Answer> Thank you for this valuable suggestion. We have now addressed the limited generalizability arising from the low prior exposure to digital therapy applications among participants (15.7%). As noted in the revised manuscript, the low familiarity with digital therapeutic tools may have influenced respondents’ perceptions and behavioral intentions, potentially limiting the extent to which the findings can be applied to populations with greater prior experience. This limitation has been added to the Discussion section.

(lines 485–491): Fourth, the generalizability of the results is limited because only 15.7% of participants had prior experience with digital therapy apps. Their limited familiarity may have shaped their attitudes and intentions differently from users who have greater exposure. Future re-search should include caregivers with more diverse levels of experience. Finally future studies should include a larger and more geographically diverse sample encompassing both urban and rural populations across South Korea to enhance the external validity of the findings.

  1. The following studies are suggested to evaluate and add to the literature review of the manuscript: https://doi.org/10.1371/journal.pone.0242023, https://doi.org/10.1016/j.ejphar.2024.176759, https://doi.org/10.3390/jtaer20030231

Answer> Thank you for your helpful suggestions. We have reviewed the recommended studies and took them into consideration during the revision process. Although they were not directly cited in the manuscript, the insights from these papers informed our refin
